# Assembly of Celastrol to Zeolitic Imidazolate Framework-8 by Coordination as a Novel Drug Delivery Strategy for Cancer Therapy

**DOI:** 10.3390/ph15091076

**Published:** 2022-08-29

**Authors:** Na Wang, Yifan Li, Fei He, Susu Liu, Yuan Liu, Jinting Peng, Jiahui Liu, Changyuan Yu, Shihui Wang

**Affiliations:** 1College of Life Science and Technology, Beijing University of Chemical Technology, Beijing 100029, China; 2Department of Gynecology, Shenzhen Traditional Chinese Medicine Hospital, Shenzhen 518033, China; 3Analytical Instrumentation Center, College of Chemistry and Molecular Engineering, Peking University, Beijing 100871, China

**Keywords:** nanomaterial, assemble, anticancer, celastrol, drug delivery

## Abstract

Celastrol (Cel), a compound derived from traditional Chinese medicine *Tripterygium wilfordii Hook.* F, has attracted considerable attention as an anticancer drug. However, its clinical application is limited due to its low bioavailability and potential toxicity. With the advancement of nanoscale metal organic frameworks (MOF), the nano-delivery of drugs can effectively improve those disadvantages. Nevertheless, hydrophobic drugs apparently cannot be encapsulated by the hydrophilic channels of MOF-based drug delivery systems. To address these issues, a new assembly strategy for hydrophobic Cel was developed by coordinating the deprotonated Cel to zeolitic imidazolate framework-8 (ZIF-8) with the assistance of triethylamine (Cel-ZIF-8). This strategy greatly elevates the assembly efficiency of Cel from less than 1% to ca. 80%. The resulted Cel-ZIF-8 remains stable in the physiological condition while dissociating and releasing Cel after a 45-minute incubation in an acidic tumor microenvironment (pH 5.5). Furthermore, Cel-ZIF-8 is proved to be easily taken up by cancer cells and exhibits a better therapeutic effect on tumor cells than free Cel. Overall, the Cel-ZIF-8 provides a novel assembly strategy for hydrophobic drugs, and the findings are envisaged to facilitate the application of Cel in cancer therapies.

## 1. Introduction

*Tripterygium wilfordii Hook.* F., also known as thunder god vine, is a traditional Chinese medicinal plant that has been used for thousands of years for its anti-inflammatory properties in the clinic [1]. In recent years, *Tripterygium wilfordii Hook.* F. has also been reported by a number of researchers to possess anticancer effects in different tumor cells such as lung cancer [2], oral cancer [3], breast cancer [4], lymphoma [5], leukemia [6], multiple myeloma [7], etc. Further, Celastrol (Cel), a naturally occurring quinone methide triterpenoid, was demonstrated to be one of the main anticancer components of *Tripterygium wilfordii Hook.* F. Cel has been proved to effectively inhibit the proliferation [8], invasion [9], and migration of various cancer cells [10]. For instance, Jing Zhang et al. reported that Cel inhibits the proliferation, invasion, and migration of human cervical cancer cells by down-regulating matrix metalloproteinase-2 (MMP-2) and MMP-9 [11]. Chun-Hau Dai et al. demonstrated that Cel and afatinib synergistically prevent the proliferation of non-small cell lung cancer cells by inducing para-apoptosis [12]. Xiaojing Li et al. have shown that Cel inhibits cancer cell proliferation and migration, induces apoptosis, and cell cycle arrest and restrains the cancer stem cell properties through suppression of Pin1 in ovarian cancer cells [10]. However, clinical applications of Cel are extremely restricted due to its strong hydrophobicity, low bioavailability, and toxicity.

To overcome the hydrophobicity and bioavailability, nanomaterials have been applied to load Cel, such as liposomes [13], protein nanoparticles (NPs) [14], mesoporous silica nanoparticles (MSN) [15], and micelles [16], whereas these delivery nanomaterials are limited to low loading capacity or unspecific release, which lowers the effectiveness and raises non-target organ toxicity of Cel [17]. Therefore, a targeted drug delivery system (DDS) with remarkable loading efficiency is urgent to be developed to provide efficient delivery of Cel for cancer treatment.

To meet this end, metal–organic frameworks (MOFs), a promising class of targeted drug carriers, are chosen to load Cel. MOFs are porous and crystalline materials formed by the self-assembly of metal ions or metal clusters with polydentate organic ligands [18,19]. The external stimuli-responsive properties, especially the acid-induced dissociation, make MOF nanocarriers one of the most widely applied targeted DDS for cancer therapy due to the pH-controlled drug release in the acidic tumor microenvironment [20,21,22]. Additionally, the tunable pore and size of MOFs allow versatile and efficient encapsulation of proteins, nucleic acids, and part of a functional molecule, enhancing the biostability in harsh conditions [23]. We firstly selected Zeolitic imidazolate frameworks 8 (ZIF-8), a famous MOF, for loading of Cel due to its high stability [24], acid response disintegration, nontoxicity, and biocompatibility [25,26,27]. ZIF-8, based on zinc ion (Zn^2+^) and 2-methylimidazole (2-MIM) linkers [28], has been applied successfully for the targeted drug delivery of doxorubicin (DOX) [29], 6-mercaptopurine (6-MP) [30], 5-fluorouracil (5-FU) [31], etc. In addition, ZIF-8 was also demonstrated to possess excellent dispersion, good homogeneity, and simple synthesis [28].

Unfortunately, after several trials, we found that the hydrophilic pore channels of the ZIF-based DDS are difficult for the direct encapsulation of hydrophobic Cel in the aqueous phase. This may be why the loading of ZIF-8 with hydrophobic drugs was rarely reported. Inspired by the deprotonable ends of Cel, i.e., the hydroxyl and carboxyl groups at two ends of the molecule, we planned to add a deprotonation agent for Cel to try to increase its solubility as well as loading efficiency. Encouragingly, we observed the deprotonated Cel assembly into a ZIF-8 frame to form a novel structure of Cel-ZIF-8 by coordinating with Zn^2+^ (Figure 1, Appendix A). This Cel-ZIF-8 has the following advantages: (1) The assembly efficiency of Cel vastly increases from <1% to ca. 80%. (2) Cel-ZIF-8 still maintains high colloidal stability in the physiological condition and degrades rapidly in acidic conditions. (3) Cel could be fast taken up by lung cancer A549 cells and exert cytotoxicity effectively by regulating the expression of apoptosis-related proteins.

In summary, in this paper, a Cel-coordinated Metal Drug Organic Framework (MdOF) delivery system is fabricated through a Cel-coordinating-based self-assembling growth strategy, which highlights its targeted delivery to acidic tumor microenvironment for cancer treatment. We believe this strategy could be a feasible solution for the assembly of hydrophobic drugs into the MdOF delivery system to improve their assembly efficiency, bioavailability, and biostability.

## 2. Results

### 2.1. Synthesis of Cel-ZIF-8 NPs

The Cel-ZIF-8 NPs were prepared by directly mixing Cel, 2-MIM, and Zn^2+^ in the presence of triethylamine (TEA). Polyvinylpyrrolidone (PVP), a hygroscopic, amorphous, biocompatible, and versatile excipient [32,33], was further added into the system to improve the biostability of Cel-ZIF-8 in aqueous solutions. The results of the Tyndall effect of Cel and Cel-ZIF-8 in the aqueous phase indicate that the free Cel is insoluble while the Cel-ZIF-8 possesses satisfactorily colloidal stability (Figure 1b). Unlike current MOF-based nanocarrier systems, which mechanically encapsulate hydrophobic molecules into their pore structures [23], the Cel-ZIF-8 are confirmed to carry Cel molecules via a definite coordination mechanism. The hydroxyl and carboxyl groups on the two ends of Cel endow it with an organic linker. The hydrogen atoms in hydroxyl and carboxyl, with the assistance of TEA, could easily be deprotonated to strengthen the coordination of Cel with Zn^2+^ (Figure 1). To identify the deprotonation effect of TEA, Cel@ZIF-8 complex in the absence of TEA was prepared for comparison. During the preparing process, most of the Cel molecules stay in the supernatant; hence, the encapsulation efficiency of Cel@ZIF-8 is lower than 1%, much lower than the assembly efficiency of Cel-ZIF-8 (~80%). This could also be verified by the much lighter color of Cel@ZIF-8 solution and powder as compared to those of Cel-ZIF-8 (Figure 1c,d).

We hypothesized that the coordinating-mediated process is initiated by coordinating deprotonated Cel with Zn^2+^ as a prenucleation cluster in the presence of TEA, followed by assembly of excess 2-MIM to the Cel-Zn^2+^ nuclei as the shell of Cel-ZIF-8. To verify this hypothesis, we initiated the process by mixing Cel and Zn^2+^ in methanol in the presence of TEA for 30 min. Within this period, the solution color gradually turns from red to dark brown, indicating the coordination of deprotonated Cel and Zn^2+^ to form the Cel-Zn^2+^ nuclei. Subsequently, excess 2-MIM was added to the solution to assemble the shell with the Cel-Zn^2+^ nuclei. The chemical structure of the deduced structure of Cel-ZIF-8 is shown in Appendix A. The solution color fades from dark brown to yellow within an hour, suggesting the successful construction of the Cel-ZIF-8 nanocomplex (Appendix A). To further confirm this hypothesis, other kinds of complexes, i.e., Cel@ZIF-8, Cel/ZIF-8, and ZIF-8, were also synthesized for comparison. Cel@ZIF-8 denotes that there is no TEA during the encapsulation. Without the assistance of TEA, Cel could not deprotonate and coordinate with Zn^2+^; thus, Cel molecules are just mechanically encapsulated into the channels ZIF-8, leading to extremely low encapsulation efficiency (1%) (Figure 1c,d). Cel/ZIF-8 means that Zn^2+^ and 2-MIM are premixed to form ZIF-8 before the addition of Cel. The color of Cel/ZIF-8 is still dark red after several hours of standing, implying that most of the free Cel molecules remain in the supernatant (Appendix A). ZIF-8, the complex without Cel, was served as the blank control (Figure 1c,d and Appendix A). Taken together, it could be concluded that Cel-Zn^2+^ nuclei should form before the assembly of 2-MIM during the fabrication of Cel-ZIF-8, which verifies our hypothesis.

### 2.2. Characterization of Cel-ZIF-8 NPs

Morphology and size of Cel-ZIF-8 were characterized by transmission electron microscope (TEM) and scanning electron microscope (SEM). The Cel-ZIF-8 NPs adopt an anticipated monodisperse rhombohedral structure with a narrow size distribution. The diameter of Cel-ZIF-8 is 57 ± 10 nm (Appendix A), which is preferred for passive tumor targeting and accumulation, presumably employing the EPR effect (Figure 1a,b). The hydrodynamic diameter of Cel-ZIF-8 detected by a Nanosizer is 72 ± 1.7 nm (Figure 1c). The size of Cel-ZIF-8 is smaller than that of ZIF-8 (107 ± 8.86 nm), stemming from the fact that the adulteration of Cel into the system limits the crystal growth (Appendix A). There was no significant difference between the Zeta potential of Cel-ZIF-8 and ZIF-8, indicating that the deprotonated Cel molecules have coordinated with Zn^2+^ and are located inside of the nanocomplex (Figure 1d).

The identity and phase purity of Cel-ZIF-8 nanocrystals were revealed by powder X-ray diffraction (PXRD) (Figure 1e). Cel-ZIF-8 nanocrystals (green line) exhibit a diffraction pattern similar to that of pure ZIF-8 (black line), and the characteristic peaks of ZIF-8 in Cel-ZIF-8 spectrum corresponding to (011), (002), (112), (022), (013) and (222) were obviously observed, indicating the minimal effect of Cel coordination on lattice distortion of ZIF-8 [34]. Moreover, the PXRD characteristic peaks of Cel in Cel-ZIF-8 nanocrystals were not observed, which agrees well with the aforementioned results that Cel serves as a coordinating agent during the assembly of Cel-ZIF-8 rather than crystallization or simply physically absorbed on the surface (Figure 1c,d, Appendix A). Moreover, in the UV-Vis absorption spectrum, Cel has a characteristic peak at 425 nm resulting from the triterpene functional group (Figure 1f). This characteristic peak was also seen in the absorption spectrum of Cel-ZIF-8, indicating that Cel still maintains its complete structure in Cel-ZIF-8. The elemental compositions of Cel-ZIF-8 and ZIF-8 were investigated through X-ray photoelectron spectroscopy (XPS). Additionally, by comparison with the XPS wide spectra of Cel-ZIF-8 and ZIF-8, it could be observed that the oxygen peak of Cel-ZIF-8 is significantly enhanced (Figure 1g). For both Cel-ZIF-8 and ZIF-8, the binding energies of Zn 2p peak at 1021.8 eV and 1044.8 eV could be assigned to Zn 2p3/2 and Zn 2p1/2 (Appendix A), and the binding energy peaks located at 530 eV and 531 eV could be ascribed to O in Zn-O and C-O-H [35]. These results indicated that Cel-ZIF-8 and ZIF-8 are constituted by the same elements. However, the peak intensity of O in Cel-ZIF-8 is much higher than that of ZIF-8, suggesting the successful coordination of the carboxylic groups in Cel with Zn^2+^ ion (Figure 1h,i). The Fourier transform infrared (FTIR) spectra of Cel-ZIF-8 and ZIF-8 are shown in Appendix A. The typical peak at 1690 cm^−1^ is assigned to the C = N stretching vibration of the imidazole ring. Peaks at 1145 and 994 cm^−1^ belong to the C–N stretching vibration. The peak at 421 cm^−1^ is ascribed to Zn-N stretching vibration. These results indicated that Cel-ZIF-8 adopts a similar structure to ZIF-8. The typical peak at 3550 cm^−1^ belongs to the carboxyl group vibration of Cel, which inferred the successful coordination of Cel into the ZIF-8 system.

### 2.3. In Vitro Drug Release and Stability of Cel-ZIF-8 NPs

Before investigating its potential application as an anticancer drug, stability, and polymer dispersity index (PDI) of Cel-ZIF-8 in PBS was first studied by using ZIF-8 as the control (Figure 2a and Appendix A). The particle sizes of Cel-ZIF-8 and ZIF-8 at pH 7.4 did not show significant changes within five consecutive days, indicating that Cel-ZIF-8 still retains the colloidal stability of ZIF-8. Therefore, we further determined whether Cel-ZIF-8 maintains the pH-responsive property of ZIF-8. As can be seen in Figure 2b, no obvious release (less than 10%) of Cel from Cel-ZIF-8 was observed after a 75-minute incubation under physiological conditions (pH 7.4), whereas more than 56% of the Cel was released after a 45-minute incubation at acidic tumor microenvironment (pH 5.5). The good dissolution behavior at pH 5.5 is helpful in improving bioavailability. These results are consistent with the pH-responsive release profile of ZIF-8 alone [25,28], indicating that the coordination of Cel does not change the pH-responsive property of ZIF-8. Since the pH-responsive property of ZIF-8 derived from the pH-sensitive of 2-MIM, it could be concluded that the coordination of Cel to Zn^2+^ does not alter the framework of ZIF-8, and most of the 2-MIM molecules assemble outside of the Cel-Zn^2+^ nuclei to form shells.

### 2.4. Cellular Uptake and Cytotoxicity Studies

The pH-responsive dissociation endows Cel-ZIF-8 with efficient tumor penetration and hypoxia relief for cancer treatment. For a drug delivery system, whether the drug can effectively enter cancer cells is of great importance. Hence the uptake of Cel-ZIF-8 by the human lung carcinoma A549 cells was determined by Laser Scanning Confocal Microscope (CLSM) using Rhodamine B (RB) and 4′,6-diamidino-2-phenylindole (DAPI) as fluorescence probes. RB could be absorbed on Cel-ZIF-8 and track the distribution of Cel-ZIF-8 in the cells by its red fluorescence (Appendix A). DAPI, a blue-fluorescent DNA stain, is utilized to characterize the location of cell nuclei. As shown in Figure 3a,b, the red fluorescence intensity increases rapidly within 2-hour uptake, indicating that A549 cells show time-dependent Cel-ZIF-8 accumulation. In other words, Cel-ZIF-8 improves the bioavailability of Cel through effective cell internalization. The cytotoxicity of Cel-ZIF-8 on A549 cells was obtained by MTT assay using ZIF-8 and free Cel (dissolved in DMSO) as comparisons (Figure 3c). No observable toxic effect was found for the ZIF-8 group because of its high biocompatibility. At the low equivalent concentration of 0.1 μM, cytotoxicity of Cel-ZIF-8 and free Cel is marginal. At an equivalent concentration of 1 μM, Cel-ZIF-8 exhibits superior cytotoxicity of 83%, which is ca. three times that of free Cel. We could reasonably suppose that the much higher cytotoxicity of Cel-ZIF-8 attributes to the better entry efficiency to tumor cells and the effective release of Cel after protonation. Taken together, Cel-ZIF-8 could be used as an effective MdOF due to its high stability in physiological condition, well acidic pH-responsive dissociation, effective cell uptake, and superior cytotoxicity.

### 2.5. In Vitro Apoptosis Assay of Cel-ZIF-8 NPs

Since Cel has been proven to be part of the framework structure of Cel-ZIF-8 (Figure 1e,h), its pro-apoptosis effect was further compared with free Cel (dissolved in DMSO). To meet this end, the effects of Cel-ZIF-8 and free Cel on the expression of apoptosis-related proteins of A549 cells were examined by Western blot (WB) experiments. By comparison with free Cel, it could be observed that the influences of Cel-ZIF-8 on the expression of caspase 3, bcl 2-Associated X (bax), and b-cell lymphoma-2 (bcl 2) do not show a significant difference (Figure 4a,b), indicating that the assembly of Cel into ZIF-8 does not change its pro-apoptotic behavior. Further, the effect of Cel-ZIF-8 dose on the apoptosis-related proteins was investigated. As shown in Figure 4c,d, with the increase in Cel-ZIF-8 dose, the expression of two pro-apoptotic proteins, i.e., caspase 3 and bax, increases, while the expression of the anti-apoptotic protein bcl 2 decreases.

## 3. Discussion

Cel is the main active component in the natural medicinal plant *Tripterygium wilfordii Hook.* F. with many physiological functions such as anti-inflammatory and anticancer. Nevertheless, the clinical application of Cel has long been limited because of its low water solubility, low bioavailability, and severe side effect stemming from the non-specific distribution. Therefore, many efforts have been made to conquer these disadvantages, mainly including chemical modification and targeted delivery. However, these methods suffer from low loading capacity, complex synthetic methods, the use of toxic organic solvents, etc. [13,36,37].

To avoid these weaknesses, we herein established a MdOF targeted delivery system for Cel using ZIF-8 as the carrier and TEA as the deprotonating agent of Cel via a simple one-pot synthesis method. During the process, the highly hydrophobic Cel molecules become deprotonated and negatively charged with the assistance of TEA. The negatively charged Cel molecules then coordinate with Zn^2+^ to form the Cel-Zn^2+^ nuclei. After that, 2-MIM molecules assemble around the nuclei to form the shell of Cel-ZIF-8. This assembly mechanism could also be confirmed by the synthesis of Cel@ZIF-8, Cel/ZIF-8, and ZIF-8 (Figure 1c,d).

To confirm the novelty of Cel-ZIF-8, we then comprehensively compared our results with those related studies in the literature. After a careful literature survey, we found that there is no study on the use of ZIF-8 or other MOFs to load Cel so far. Therefore, this is the first time to use ZIF-8 as a nanocarrier for the targeted delivery of Cel. We then compared our results with other loading methods for Cel delivery, such as micelles, protein NPs, MSN, and liposomes [13,14,15,16]. Details as below.

(1)For micelles carriers [16], this nanomaterial is untargeted. Moreover, the loading capacity of Cel is ~5%, which is significantly lower than that reported in our work (~80%).(2)For MSN [15], the loading capacity of Cel is ~26%, which is also greatly lower than that of Cel-ZIF-8 (~80%). Additionally, the synthesis was conducted under a high temperature of 80 °C, while our Cel-ZIF-8 could swiftly assemble under room temperature.(3)For protein NPs [14], the loading capacity of Cel was not mentioned in the study. The synthesis steps of nanoparticles involved a longer time of ~3–4 days, and toxic organic solvents such as dichloromethane were utilized during the process. Nevertheless, our Cel-ZIF-8 was synthesized by the self-assembly of Cel, Zn^2+^, and 2-MIM via a simple one-pot method, which only involves about ~2 h. Additionally, the synthesis process was also environmentally friendly, for no toxic organic solvents were used in the synthesis.(4)For liposome carriers [13], this nanomaterial is untargeted, and the loading capacity of Cel was also unavailable. The synthesis involved an incubation time of 36 h, which is much longer than that of Cel-ZIF-8 (~2 h). In addition, toxic solvents such as chloroform were used during the synthesis, which is harmful to the environment.

In summary, the Cel-ZIF-8 constructed in this work possesses three main advantages:(1)Cel-ZIF-8 sharply improves the aqueous solubility of Cel by deprotonating its hydrogen atoms in hydroxyl and carboxyl, which vastly elevates the assembly efficacy from <1% to 80%.(2)Cel-ZIF-8 greatly elevates the bioavailability and cytotoxicity of Cel. In Section 2.4 and 2.5 (Figure 3 and Figure 4), the cellular uptake, cytotoxicity, and pro-apoptosis capabilities of Cel-ZIF-8 and Cel were comprehensively compared. Although the results were mostly similar, it should be noticed that the free Cel was actually dissolved in DMSO instead of an aqueous solution due to its low water solubility. In other words, the actual bioavailability of Cel is extremely low under physiological conditions. Therefore, it could be concluded that the Cel-ZIF-8 greatly improves the bioavailability of Cel.(3)Cel-ZIF-8 switches the bio-distribution of Cel from non-specific to pH-responsive targeted delivery, thus lowering the side effect of Cel. As depicted in Section 2.3, Cel-ZIF-8 maintains colloidal stability under physiological conditions (pH 7.4) while dissociating an acidic tumor microenvironment (pH 5.5). Additionally, it could be deduced from Figure 3a that RB molecules were loaded in Cel-ZIF-8, which enlightens us that another small and hydrophilic drug could also be loaded in Cel-ZIF-8 as a synergetic component of Cel.

Additionally, the synthesis of Cel-ZIF-8 is simple, environmentally friendly, and time-saving. Therefore, the Cel-ZIF-8, as a Cel-coordinated MdOF targeted delivery system, is a promising strategy for Cel targeted efficiency delivery and provides an insight into the loading of hydrophobic drugs on MOFs. It is expected that Cel-ZIF-8 will be applied to many cancer treatments in the near future.

## 4. Materials and Methods

### 4.1. Materials and Reagents

Celastrol (99.4%) was purchased from Shanghai Bide Medical Technology Co., Ltd. (Shanghai, China). Zinc nitrate hexahydrate (Zn(NO_3_)_2_·6H_2_O, AR, 98%), 2-methylimidazole (2-MIM, RG, 98%), thiazolyl blue tetrazolium bromide (MTT, 98%), polyvinyl pyrrolidone (PVP), Rhodamine B (RB), and anhydrous methanol were bought from Beijing Innochem Science & Technology Co., Ltd. (Beijing, China). Triethylamine (TEA) ((C_2_H_5_)_3_N, AR, 99%) was acquired from Xilong Chemical Co., Ltd. (Shantou, China). RPMI medium, Western blot primary antibody against bax, Caspase 3, bcl-2, β-Actin, and goat anti-rabbit/mouse IgG secondary antibody were purchased from Proteintech (Chicago, IL, USA). Fetal bovine serum (FBS), penicillin/streptomycin (PS), the cell lysis buffer, Annexin V-FITC Apoptosis Detection Kit, and DAPI staining solutions (10 μg/mL) were obtained from Beyotime Biotechnology (Shanghai, China). All other chemicals were of the highest purity available from local sources and used without further purification.

### 4.2. Synthesis of Cel-ZIF-8, Cel@ZIF-8, Cel/ZIF-8, and ZIF-8 Nanoparticles (NPs)

For the synthesis of Cel-ZIF-8, an aliquot of 200 μL Cel (10 mg/mL) was mixed with 7.4 mL methanol and 0.1 mL TEA with stirring. The sample was dropped with 0.3 mL Zn(NO_3_)_2_·6H_2_O methanol solution (125 mg/mL) (termed as Zn^2+^ hereafter) and stirred vigorously for 30 min until the color changed from yellow to clear reddish-brown. The solution was added with 100 mg PVP with stirring until PVP was totally dissolved. Then, 2 mL 2-MIM methanol solution (25 mg/mL) was dropped into the solution and reacted for 1 h till the color became lighter. The product was centrifuged for 20 min under 10,000 rpm. The supernatant was washed three times with deionized water to obtain Cel-ZIF-8. The Cel-ZIF-8 was added with deionized water to adjust the concentration to 25 mg/mL and stored at 4 °C for further analysis. Cel@ZIF-8 NPs were synthesized by mixing Cel, Zn^2+^, and 2-MIM in methanol solution and stirring for 1 h. The concentrations and doses of the solutions were consistent with those in the synthesis of Cel-ZIF-8. The ZIF-8 NPs were also prepared for control by replacing Cel solution with methanol. Cel/ZIF-8 was synthesized by stirring free Cel and ZIF-8 NPs in methanol solution with stirring for 24 h and centrifuging at 10,000 rpm for 10 min.

### 4.3. Determination of Cel Assembly Efficiency

To determine Cel content in Cel-ZIF-8 NPs, the following formula was used to calculate the assembly efficiency (also referred to as coordination amount in some literature).
(1)$$Assembly\;efficiency=\frac{[{\mathrm{Cel}}_{\mathrm{add}}]-[{\mathrm{Cel}}_{\sup}]}{\left[{\mathrm{Cel}}_{\mathrm{add}}\right]}\times 100\%$$

Among them, [Cel_add_] represents the total amount of Cel and [Cel_sup_] represents the Cel in the supernatant after synthesis.

### 4.4. Characterization of the ZIF-8 and Cel-ZIF-8 NPs

The morphology and size of Cel-ZIF-8 and ZIF-8 NPs were characterized by a scanning electron microscope (SEM, MAIA3, TESCAN, Shanghai, China) and a transmission electron microscope (TEM, FEI Talos F200X, Thermo Fisher Scientific, Waltham, MA, USA). The particle sizes and Zeta potentials of Cel-ZIF-8 and ZIF-8 NPs were obtained by a Malvern Zetasizer Nano ZS90 (Malvern Instruments, Malvern, UK). Characterization and structure confirmation of Cel-ZIF-8 and ZIF-8 was performed by a Fourier transform infrared spectroscope (FTIR, VERTEX 70v, Bruker, Saarbrücken, Germany) within the range of 4000−400 cm^−1^. The X-ray diffraction pattern was detected by a BRUKER D8 ADVANCE diffractometer (Bruker, Saarbrücken, Germany) at the scanning speed of 0.02 s^−1^ in the range of 2θ recorded from 5° to 40°. The copper target X-ray tube was set as 40 kV and 40 mA employing Cu Kα radiation to obtain powder X-ray diffraction (PXRD) patterns. The X-ray photoelectron spectroscopy (XPS) data were achieved with an Axis Supra electron spectrometer from SHIMADZU using 156 W Al kα radiation. UV/Vis’s absorption spectra were analyzed using a NanoDrop spectrophotometer (Thermo Fisher Scientific, Waltham, MA, USA).

### 4.5. Cell Culture and Cytotoxicity Test

The human lung carcinoma A549 cell line was used to assess the biosafety of Cel-ZIF-8. Briefly, A549 cells were cultured in the RPMI medium supplemented with 10% FBS and 100 U/mL PS in a humidified incubator (MCO-18AIC, SANYO) at 37 °C with 5% CO_2_. Cells were seeded into a 96-well plate with 1 × 10^4^ cells/well and incubated for 24 h. The 96-well plate medium was discarded and washed with PBS. Then medium containing different concentrations of Cel-ZIF-8 ranging from 0.045 to 4.5 μg/mL was added to each well and incubated for 24 h. Cell viability was determined using an MTT assay according to a standard protocol.

### 4.6. Cellular Uptake and Confocal Laser Scanning Microscopy (CLSM) Analysis

Cellular uptake of Cel-ZIF-8 by A549 cells was determined by CLSM employing RB and DAPI as fluorescence probes according to a previously described method [38]. An aliquot of 2 mL A549 cells (5 × 10^4^ cells/mL) in RPMI medium was incubated in laser confocal dishes at 37 °C for 24 h to allow adherence. Then, the culture medium was replaced by a 2 mL fresh medium containing 0.25 mg Cel-ZIF-8 and 20 μg RB. After incubation for 0.5, 1, or 2 h, the medium was replaced by 100 μL 10 μg/mL DAPI and cultured for 10 min in the absence of light. Images were captured by exciting DAPI at 358 nm and RB at 554 nm and measuring the emitted light at 461 and 625 nm for DAPI and RB, respectively. Images of samples were collected by a Leica SP8 inverted confocal microscope (Leica Microsystems, Wetzlar, Germany).

### 4.7. Detection of Apoptosis-Related Proteins by Western Blot

Western blot analysis of Cel-ZIF-8 treated cells was performed as previously reported [39]. The A549 Cells (1 × 10^5^ cells/well) were co-incubated with 5 μM Cel-ZIF-8 (equivalent to 20 μg/mL ZIF-8), 5 μM Cel, or 20 μg/mL ZIF-8 in a 6-well plate for 48 h. Then, the cells were lysed with a Cell lysis buffer and centrifuged at 5000 rpm and 4 °C for 5 min. The total protein concentration was determined by a BCA Protein Assay Kit and adjusted to 50 mg/mL. Subsequently, the proteins were added with a 5×SDS loading buffer and denatured in a dry bath incubator (MK-20, ALLSHENG) at 95 °C. The denatured proteins were separated by a 15% SDS-PAGE, transferred onto a PVDF membrane in an ice bath under 350 mA for 100 min, and blocked using 8% skimmed milk powder. After that, the proteins were co-incubated with the primary antibody at 4 °C overnight. After washing, proteins were added with the secondary antibody and incubated at room temperature for 1 h. Finally, proteins were imaged under a chemiluminescence apparatus (CLINX, China).

## 5. Conclusions

In summary, we developed a promising therapeutic alternative nanoplatform Cel-ZIF-8 for cancer treatment. The size of Cel-ZIF-8 is 57 ± 10 nm, and the Zeta potential is ca. 25 mV. Different from the mechanical encapsulation of drugs, the deprotonated Cel performs as a coordinating agent to nucleate with Zn^2+^ and then assemble with 2-MIM to form Cel-ZIF-8. This greatly elevates assembly efficiency from less than 1% to ca. 80%. Meanwhile, the incorporation of Cel maintains the stability in the physiological condition and the low pH-responsive property of the ZIF-8 framework. Cel-ZIF-8 can be easily taken up by cancer cells within 2 h and exert 83% of cytotoxicity, which is ca. three times that of free Cel. Therefore, the water-soluble MdOF delivery platform Cel-ZIF-8 offers a novel assembly strategy for hydrophobic drugs and is envisaged to facilitate the development of cancer treatment.

## Data Availability

Data is contained within the article and Appendix A.

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
