# Peer review of "Assembly of Celastrol to Zeolitic Imidazolate Framework-8 by Coordination as a Novel Drug Delivery Strategy for Cancer Therapy"

_pharmaceuticals, 2022, doi:10.3390/ph15091076_

Round 1
Reviewer 1 Report
In this paper, the authors reported the preparation of celastrol (Cel) – zeolitic imidazolate framework-8 (ZIF) complex compound (Cel-ZIF-8) by self-assembling synthesis method as drug delivery for cancer therapy.
The synthesis method is simple but useful to make nano-delivery for hydrophobic drugs.
They also reported that the Cel-ZIF-8 works well as drug delivery in vitro.
The characterization of Cel-ZIF-8 was performed suitably by various measurements such as SEM, TEM, FTIR, PXRD, XPS and UV/Vis absorption spectra.
The drug release and stability of Cel-ZIF-8 in vitro were investigated carefully. And cellular uptake, cytotoxicity and apoptosis were also studied.
I think that the results reported in this paper are useful for drug delivery for cancer therapy and contribute to develop the field of drug delivery.
There are some minor comments as follows.
1. Scheme 1b
CE. → Cel ?
2. P. 7, line 14
Fig. 3 and 4 → Fig. 3 ?
3. I have a question for the structure of Cel-ZIF-8. If the simple figure of the deduced structure of Cel-ZIF-8 was shown in the paper or supplement, I could easily understand the structure.
Reviewer 2 Report
The overall process of deriving research results is smooth without any problems. However, most importantly, a rationale and basic experimentation for the reason for introducing this system is required. To this end, we suggest that it is necessary to further highlight the scientific significance of this study through the following basic additional revision.
1. Is ZIF-8 with the same structure applicable? Is there any reason to choose ZIF 67 over ZIF-8?
2. What is the bond between Celastrol (Cel)/ZIF 67, physical bond or chemical bond? If it is a chemical bond, author should show with evidence how it is done specifically.
3. How much is the amount of coordinating the deprotonated Cel to ZIF? The authors need to quantify this and should insert these data in the manuscript.
4. ZIF-8 composite materials for various application are reported, and the latest paper as follows should be updated and cited.:
(1) Rapid adsorption and removal of sulfur mustard with zeolitic imidazolate frameworks ZIF-8 and ZIF-67 YR Son, SG Ryu, HS Kim Microporous and Mesoporous Materials 293, 109819
Reviewer 3 Report
My deep thanks for you to review the following manuscript : Assembly of celastrol to zeolitic imidazolate framework-8 (ZIF8) by coordination as a novel drug delivery strategy for cancer therapy.
The submitted work is introducing a new valuable and interesting idea and the given results confirm the idea. This work is suitable for publication in the Journal. I suggest the acceptance after some major corrections as follows;
1. Abstract section need to rewrite in correct sequence
2. Enhance the introduction section because it was very thin
3. Compare your results by literature survey
4. FTIR need to be added
5. What is the application of this study ?
6. Reformulate the aim of the work in introduction
7. Add previous published work with comparison to clear the novelty of your work
8. Give some results with numbers in conclusion
9. The caption of all figures is very thin, need to add more details in each caption
10. Add more explanation to experimental work
11. The bibliography needs to be improved. Some papers in literature could be taken into consideration such as; Wound dressing properties of functionalized environmentally biopolymer loaded with selenium nanoparticles -- It is good to mention and add all these articles that could be important in the introduction section and add to references section
12. Correct typographical errors.
13. Don’t use abbreviations in title and abstract, you must define it in first time use
Round 2
Reviewer 2 Report
I'm satisfied with revised manuscript.
I recommend that this manuscript be published.
Reviewer 3 Report
accept the paper in this form